# The well-being and work-related stress of senior school leaders in Wales and Northern Ireland during COVID-19 *"educational leadership crisis"*: A cross-sectional descriptive study

Emily Marchant[1]*, Joanna Dowd[2], Lucy Bray[3], Gill Rowlands[4], Nia Miles[5], Tom Crick[1], Michaela James[6], Kevin Dadaczynski[7,8], Orkan Okan[9,10]

1 Department of Education and Childhood Studies, Faculty of Humanities and Social Sciences, Swansea University, Swansea, United Kingdom, 2 Health Researcher (Freelance), Northern Ireland, United Kingdom, 3 School of Nursing, Midwifery and Allied Health, Faculty of Health, Social Care and Medicine, Edge Hill University, Edge Hill, United Kingdom, 4 Public Health Sciences Institute, Newcastle University, Newcastle, United Kingdom, 5 National Academy for Educational Leadership Wales, United Kingdom, 6 National Centre for Population Health and Wellbeing Research, Population Data Science, Faculty of Medicine, Health and Life Science, Swansea University, Swansea, United Kingdom, 7 Department of Health Sciences, Fulda University of Applied Sciences, Fulda, Germany, 8 Centre for Applied Health Sciences, Leuphana University Lüneburg, Lüneburg, Germany, 9 Department of Sport and Health Sciences, Center for Health and Medicine in Society, School of Medicine and Health, Munich, Germany, 10 Department of Sport and Health Sciences, Center for Health Promotion in Childhood and Adolescence, School of Medicine and Health, Munich, Germany

☯ These authors contributed equally to this work.
* E.K.Marchant@swansea.ac.uk

**Data Availability Statement:** Data cannot be shared publicly because of ethical requirements regarding confidentiality, for which participants did

## Abstract

The COVID-19 pandemic caused far-reaching societal changes, including significant educational impacts affecting over 1.6 billion pupils and 100 million education practitioners globally. Senior school leaders were at the forefront and were exposed to particularly high demands during a period of "*crisis leadership*". This occupation were already reporting high work-related stress and large numbers leaving the profession preceding COVID-19. This cross-sectional descriptive study through the international COVID-Health Literacy network aimed to examine the well-being and work-related stress of senior school leaders (n = 323) in Wales (n = 172) and Northern Ireland (n = 151) during COVID-19 (2021–2022). Findings suggest that senior school leaders reported high workloads (54.22±11.30 hours/week), low well-being (65.2% n = 202, mean *WHO-5* 40.85±21.57), depressive symptoms (*WHO-5* 34.8% n = 108) and high work-related stress (PSS-10: 29.91±4.92). High exhaustion (BAT: high/very high 89.0% n = 285) and specific psychosomatic complaints (experiencing muscle pain 48.2% n = 151) were also reported, and females had statistically higher outcomes in these areas. School leaders were engaging in self-endangering working behaviours; 74.7% (n = 239) gave up leisure activities in favour of work and 63.4% (n = 202) sacrificed sufficient sleep, which was statistically higher for females. These findings are concerning given that the UK is currently experiencing a "*crisis*" in educational leadership against a backdrop of pandemic-related pressures. Senior leaders' high attrition rates further exacerbate this,

not consent to. Data are available from the Swansea University Medical School Research Ethics Committee (contact via FMHLS-Ethics@swansea.ac.uk) for researchers who meet the criteria for access to confidential data.

**Funding:** The Economic and Social Research Council (ESRC) funded EM (grant number: ES/W007045/1) and the development of the HAPPEN network (grant number: ES/J500197/1) which this research was conducted through (https://esrc.ukri.org/). The funders had no role in study design, data collection and analysis, decision to publish, or preparation of the manuscript.

**Competing interests:** The authors have declared that no competing interests exist.

proving costly to educational systems and placing additional financial and other pressures on educational settings and policy response. This has implications for senior leaders and pupil-level outcomes including health, well-being and educational attainment, requiring urgent tailored and targeted support from the education and health sectors. This is particularly pertinent for Wales and Northern Ireland as devolved nations in the UK, who are both implementing or contemplating major education system level reforms, including new statutory national curricula, requiring significant leadership, engagement and ownership from the education profession.

## Introduction

The COVID-19 pandemic caused by the transmission of severe acute respiratory syndrome coronavirus 2 (SARS-CoV-2) resulted in unprecedented societal changes. In response, a range of public health measures were implemented to reduce social contacts and transmission of SARS-CoV-2. This included changes to the delivery of learning, teaching and assessment within educational settings and contexts. Measures included full or partial closure of face-to-face provision, a move to hybrid and blended learning, and the introduction of a variety of school-based measures upon the full return to education [1–6]. These significant and prolonged changes to the delivery of teaching and learning impacted all layers of education, including pupils, educational support staff, teachers, and senior school leaders (e.g. headteachers, deputy headteachers, senior leadership team) [7]. At its peak, over 1.6 billion learners and 100 million educational practitioners globally were affected, and the pandemic has been deemed the most unprecedented disruption in the history of education [8].

Research efforts were initially focused on the impacts of the pandemic on school children and teaching staff [3, 6, 8]. However, senior school leaders were at the forefront of educational leadership and were required to manage a shift in traditional leadership roles and navigate an evolving working situation and environment [9]. This required rapid decision-making and alterations to school management and leadership relating to the numerous new protocols and policies they had to master and manage. Responsibilities included rapidly responding to government guidelines, managing communication with education authorities and the school community, ensuring professional development and continuity of learning and safeguarding vulnerable children. Senior school leaders are responsible for all aspects of school life and therefore had to cope with high demands due to the COVID-19 pandemic, a period defined as "*crisis leadership*" [10, 11]. This study aimed to examine the working situation, well-being and work-related stress of senior school leaders in Wales and Northern Ireland during this period of crisis leadership.

The term crisis leadership is characterised as an urgent situation requiring immediate and decisive action by an organisation or leaders of the organisation [12]. In the context of education, evidence regarding crisis leadership is limited but explores how educational leaders manage a period of "*unexpected, fundamental disruption to school functioning with potentially high consequences for the organization, its stakeholders, and its reputation*" [13, p.315]. To date, scoping reviews demonstrate the majority of research focuses on the response to and recovery from natural disasters and human-made crises, with a significant lack of research on educational leadership during public health crises [14]. However, crisis leadership in education was brought to the forefront during COVID-19, and a growing body of evidence has drawn attention to its importance in securing stability for learners,

parents, staff and the community during a time of unprecedented disruption [9, 12, 13, 15]. Recent work has focussed on developing frameworks to understand the processes behind the management of educational crises, such as the five phase cycle of *mitigation/prevention*, *preparednes*s, *response*, *recovery* and *learning* [13]. Whilst our understanding of the conditions that enable effective educational crisis leadership is growing [13, 14], these frameworks focus on the skills and attributes of leaders during the process of crisis leadership, neglecting outcomes such as how school leaders deal with and are impacted by this period. Thus, there is a gap in empirical research examining the impact on how school leaders experienced crisis leadership during COVID-19. This is important because factors such as well-being and work-related stress may be impacted which are fundamental during and after periods of crisis leadership.

Further, concerns were already being raised regarding educational leaders' workload, well-being, recruitment and retention within the profession prior to COVID-19 [16, 17], and experts had referred to a potential crisis in leadership in education [18]. It is well documented that educational leaders are prone to high levels of work-related stress, burnout, and reduced well-being, reporting higher stress levels than other occupations and the general population [19–22]. Workload and work-related stress have been cited as key reasons for leaving the profession [16, 23] and between 2020 and 2022 there was an 82% increase in all teaching staff leaving the profession in Wales [24]. Furthermore, evidence suggests gender differences exist in perceptions of stress and exhaustion, with female school leaders exhibiting higher perceived stress [25] and symptoms of exhaustion and fatigue compared to their male counterparts [26]; further research is required to examine this in the context of COVID-19.

These existing pressures on educational leaders were exacerbated during the COVID-19 pandemic in a time that required "*constant crisis and change management*" [27]. This was fuelled further by intense scrutiny of school leaders by policymakers, parents and the media, in addition to a shift from local to national decision making and accountability [28]. A report by the National Association of Head Teachers (NAHT) that surveyed over 2,000 senior leaders across England, Wales and Northern Ireland in 2021 found increasing dissatisfaction of school leadership as a career choice. Almost half of respondents did not recommend school leadership and over half of those in deputy/assistant leadership roles reported to not aspire to progress to senior headship level [23].

These psychological and physical impacts can transcend to learners, with a body of evidence demonstrating an association between school leadership quality and student outcomes, including achievement, health and well-being [29, 30]. The additional work-related pressures during a period of "*crisis leadership*" are likely to have had significant impacts on educational leaders, and research is urgently required to assess these impacts to direct appropriate resources and support.

This cross-sectional descriptive study was conducted through the international *COVID Health Literacy* (COVID-HL) network [31]. The network was established in 2020 to enable collaborative and cross-country health literacy research and includes more than 150 researchers from over 60 countries. Through the COVID-HL network, a number of surveys were developed to examine the impact of the COVID-19 pandemic on educational settings, including the *COVID-19 School Principals Survey* [32], which has been administered in 17 countries to date [33–37]. This study aimed to examine the working situation, well-being and work-related stress of senior school leaders in Wales and Northern Ireland and explore gender differences as a whole sample in a period of educational crisis leadership during the COVID-19 pandemic 2021 and 2022.

## Materials and methods

The *COVID-19 School Principals Survey* was administered to senior school leaders, recognised in this study as headteachers, deputy/assistant headteachers, or school staff with leadership and/or management responsibilities. The survey was conducted in Wales (n = 172) between June and November 2021 and Northern Ireland (n = 151) between March and May 2022. This was part of an ongoing international study through the COVID-HL network, and the survey has been administered in 17 countries globally at the time of writing.

### Context

Education and health are devolved responsibilities of the respective governments of Wales and Northern Ireland within the UK. There was some general consistency in both countries' educational response to COVID-19 including periods of closure and the reopening of face-to-face educational provision, blended and hybrid learning, and the implementation of school-based measures (e.g. social distancing) [38, 39]. However, there was variation in the timings of implementation and relaxation of measures during the entirety of the COVID-19 pandemic and within the study period [38, 39].

In Wales, the collection of data shortly followed the re-opening of face-to-face education (February-April 2021), and encompassed requirements of self-isolation and hybrid learning (June and July 2021), teacher-based assessments (June 2021), increases in COVID-19 cases linked to schools (July and October 2021), regular lateral flow testing of pupils (secondary) and teachers (September 2021) and the reintroduction of school-based public health measures (wearing face coverings). In addition to this, schools across Wales were preparing to roll out a new curriculum, the *Curriculum for Wales* [40] following significant education reform [41]. In Northern Ireland, the data collection period was later into the pandemic. Whilst education had returned somewhat to pre-pandemic delivery, this was amongst a backdrop of continued high COVID-19 rates, significant staff and pupil absences (January and February 2022), and ongoing COVID-related impacts and sustained pressures on the education system [39].

### Conceptual framework

This study is conceptually positioned within the framing of educational crisis leadership. Defined above, this study recognises this as a period of "*unexpected, fundamental disruption to school functioning with potentially high consequences for the organization, its stakeholders, and its reputation*" [13, p.315] that requires immediate and decisive action by school leaders [12]. Whilst this study does not aim to examine the conditions and factors associated with effective educational crisis leadership, it aims to examine how school leaders experienced crisis leadership and the perceived impact this had on their working situation, well-being and work-related stress. These are currently neglected areas of study with the literature. In recognition of current frameworks on educational crisis leadership [13], this study informs the *recovery* and *learning* phases of these models based on school leaders' experiences.

### Sample and recruitment

Inclusion criteria were: any member of staff with a senior leadership position, recognised in the UK as senior leadership team (SLT). This includes the headteacher or deputy/assistant headteacher, in addition to members of the SLT with leadership or management responsibilities (e.g. head of school/department/subject area, or senior pastoral role such as pupil well-being). Senior leaders were required to be currently working within any primary (ages 3–11), secondary (ages 11–16) or special educational setting. A convenience sample of participants

(total: n = 323, Wales: n = 172, Northern Ireland: n = 151) were recruited via email, social media and through education stakeholders. Recruitment in Wales was facilitated by HAPPEN (**H**ealth and **A**ttainment of **P**upils in **P**rimary **E**ducatio**N**) Wales [42, 43], a pan-Wales infrastructure that connects research with primary schools, directly aligning with the new *Curriculum for Wales* [44] (which is phasing in from September 2022) and its health and well-being area of learning and experience, alongside the increased policy prominence of health literacy. The recruitment period in Wales was from 1st June 2021 to 14th November 2021. In addition, stakeholder support was received from the National Academy for Educational Leadership Wales [45] who distributed the survey through their networks. In Northern Ireland, the survey was emailed to all primary, post-primary and special schools, and a reminder was also sent to encourage uptake. The recruitment period in Northern Ireland was from 31st January 2022 to 31st May 2022. The survey was securely administered online via Microsoft Forms with a survey link emailed to participants. Participants received an information sheet detailing the study aims, objectives, their rights, including their right to withdraw and information regarding anonymity and confidentiality and were required to provide written informed consent, confirming that they had read the information sheet and understood what participation involved. Additional information regarding the ethical, cultural, and scientific considerations specific to inclusivity in global research is included in the S1 Checklist.

## The *COVID-19 School Principals Survey*

The *COVID-19 School Principals Survey* [32] was designed by the international COVID-HL network and contained 33 questions. These included standardized scales or adaptations to existing surveys to the COVID-19 pandemic. The original survey consisted of various categories and sub-categories; socio-demographic information (e.g. gender, type of school), current work situation (e.g. Sense of Coherence, perceived stress), health information in the context of COVID-19 (e.g. health literacy), health promotion and prevention in school (e.g. health promoting school factors) and health situation (general health, well-being). A full breakdown of categories and sub-categories is presented in S1 Fig. For the purpose of this study, questions relating to work situation, health, well-being, work-related Sense of Coherence, exhaustion and psychosomatic complaints, self-endangering behaviour and perceived stress were used for analyses. The survey was adapted for Wales and Northern Ireland including country specific wording, additional questions seeking demographic characteristics, and minor re-structuring (S2 and S3 Figs). The inclusion criteria for the survey were any senior leadership staff (headteacher, deputy headteacher, senior leadership team). The survey was completed on Microsoft Forms and took an average completion time of 30 minutes for participants in Wales and 28 minutes for participants in Northern Ireland. A full copy of the survey for Wales and Northern Ireland is available in the S2 and S3 Figs. A full copy of the original survey is available upon request.

## Measures

The 33-item *COVID-19 School Principals Survey* captured demographic characteristics in addition to the integration of adapted or standardized scales. These included work-related factors (weekly workload/teaching load), general health, well-being (WHO-5), work-related Sense of Coherence (SoC), exhaustion and psychosomatic complaints (short-form items from the *Burnout Assessment Tool* (BAT)), self-endangering behaviour (self-endangering work behaviour scale) and perceived stress (*Perceived Stress Scale*).

**Demographic characteristics.** Data regarding demographic characteristics were collected, including gender (male, female, prefer not to say), type of school (primary, post-

primary/secondary), specific leadership position at school (e.g. headteacher, deputy headteacher, senior leadership team e.g. with leadership, management or pastoral responsibilities), school size, percentage of pupils from different socioeconomic groups (self-defined as low, medium, high) and percentage of their pupils eligible for free school meals. Free school meals are used as a proxy measure of deprivation, eligibility criteria at the time of the study were any child living in a household which gets income-related benefits and has an annual income less than £7,400 in Wales [46] or £14,000 in Northern Ireland [47].

**Work-related factors.**    Participants were asked questions regarding their workload, including their total weekly working hours, weekly teaching hours and changes in their workload compared to before the COVID-19 pandemic (higher, about the same, lower than before the COVID-19 pandemic).

**General health.**    The subjective health of participants was examined using the WHO-endorsed item of the self-perception of health status, which asks about '*health in general*' [48]. This has been found to be associated with other health measures, use of health services and survival rate in adults. In the current study, participants responded to a single-item question '*How is your health in general*?' using a 5-point Likert scale (1–5) from '*Very good*' to '*Very bad*' [49]. Lower mean values indicate better perceived general health.

**Well-being.**    The 5-item *World Health Organization Well-being Index* (*WHO-5*) was used to measure the subjective psychological well-being of senior school leaders [50]. Participants were asked to indicate how they had been feeling in the past two weeks in relation to five statements, including '*I have felt cheerful and in good spirits*'. Responses followed a 6-point Likert scale and were scored from 0 (*None of the time*) to 5 (*All of the time*). These were summed to calculate a total raw score between 0 to 25, and this total raw score was multiplied by 4, providing a final *WHO-5* score between 0 (complete absence of well-being) to 100 (highest imaginable level of well-being). Mean scores were calculated, in addition to binary variables, a cut-off score of ≤50 is indicative of low well-being [51], and ≤28 indicative of depressive symptoms [52].

**Work-related Sense of Coherence.**    *Sense of Coherence* (SoC) is a framework which explains how people manage stressful situations to maintain their health and well-being. Work-related SoC is used as an indicator for the health-promoting quality of life at work. It consists of three concepts; comprehensibility (how an individual perceives their work situation as structured, consistent and coherent, as opposed to unpredictable and chaotic), manageability (how an individual perceives the availability of resources to cope with demands in the workplace) and meaningfulness (the extent to which a person perceives their work situation as worthy of commitment and involvement) [53]. Work-related SoC was captured using a 9-item scale with responses following a Likert scale (each item score ranging from 1–7) relating to how participants were finding their work situation. For the comprehensibility sub-scale (items 1, 3, 6 and 9), items were '*Unmanageable–manageable*', '*Unstructured–unstructured*', '*Unclear–clear*' and '*Unpredictable–predictable*'. The manageability sub-scale (items 4 and 7) includes '*Impossible to influence–easy to influence*' and '*Uncontrollable–controllable*'. The meaningfulness sub-scale was captured in items '*Meaningless–meaningful*', '*Insignificant–significant*'. Total scores were summed, and the three sub-scales scores were summed, and mean values calculated. Higher values indicate a higher SoC.

**Exhaustion and psychosomatic complaints.**    Work-related exhaustion was assessed using the short-form exhaustion items from the *Burnout Assessment Tool* (BAT) [54]. This asked participants to rate on a 5-item Likert scale from Never to Always (scored 1–5) three statements relating to how they were experiencing their work situation including '*At work, I feel mentally exhausted*'. For psychosomatic complaints, the following items from the BAT were assessed using the same Likert scale as above, asking participants how often they suffer from, for example, '*Palpitations and/or chest pain*'. Total scores and mean values for exhaustion and

psychosomatic complaints were calculated, with higher scores indicating higher work-related fatigue or psychosomatic complaints. In addition, statistical norms based on percentiles derived from Schaufeli, De Witte and Desart [54] categorised the exhaustion sub-scale as low, average, high or very high.

**Self-endangering behaviour.** Participants' self-endangering behaviour, recognised as behaviours that may be functional to attaining work goals in response to coping with excessive working demands, but have a negative effect on longer-term health, well-being and ability to work were assessed. Three sub-scales of the subjective self-endangering work behaviour scale were used; the 6-item work extensification (extending working hours), 3-item work intensification (working at an increased pace and multitasking, whilst limiting break periods and work-related social interactions) and 3-item quality reduction (reducing the quality of work in reaction to excessive work demands). For each sub-scale, participants were asked in relation to the past three months, and responses were scored using a 5-point Likert scale from '*Never*' to '*Very often*'. Total scores and mean values were calculated for each sub-scale, with higher values indicating higher self-endangering behaviours. Krause et al. [55] report very good reliability of the subjective self-endangering work behaviour scale and acceptable to good reliability within sub-scales.

**Perceived stress.** Perceived stress was captured using a 10-item measure based on the original *Perceived Stress Scale* by Cohen [56], adapted by Schneider et al. [57] and applied to the COVID-19 working context by Dadaczynski, Okan and Messer [32]. Participants were asked how often they had found aspects of their working situation in the last month, including '*Felt confident about your ability to handle your professional work-related problems caused by the COVID-19 pandemic*'. The 10-item adapted PSS consists of two sub-scales; *Perceived Helplessness* (items 1, 2, 3, 6, 9, 10), which represents a lack of control of negative emotions, and *Perceived Self-Efficacy* (items 4, 5, 7, 8), how effective the person feels in coping with demands. A 5-item Likert scale scored responses were from 1 ('*Never*') to 5 ('*Very often*'). A total score was summed by combining perceived helplessness item scores and the reverse scoring of perceived self-efficacy items. In addition, sub-scale total scores (perceived self-efficacy reverse scored) were calculated. Total mean and sub-scale values were calculated, higher values indicate higher perceived stress. High content and construct validity have been reported by Schneider et al. [57].

## Data analysis

Raw data were downloaded from Microsoft Forms, cleaned and checked for duplicates, and unique participant ID numbers generated. Data were then handled using IBM SPSS Statistics (version 28.0 for Mac). Raw survey responses were coded following COVID-HL study documentation protocol, this ensures consistency across international study data. Next, descriptive statistics were calculated describing frequencies and percentages, mean values and standard deviations for the variables listed above. This was done in three phases, as a whole sample, by country and by gender. Independent samples t-tests explored differences between gender, with the level of statistical significance set as a two-sided $p < 0.05$.

## Ethics

This study received ethical approval from the Swansea University Medical School Research Ethics Committee (approval number: 2021–0043). All participants were required to provide written informed consent to participate in the study. Participants in Northern Ireland were made aware that due to the anonymous nature of the survey, they were not able to withdraw

their data after pressing submit. For Wales, personal data (name and school) were collected for the purposes of data linkage, and participants were informed of their right to withdraw.

## Results

Findings from this study suggest that senior school leaders reported high workloads (54.22 ±11.30 hours/week), low well-being (65.2% n = 202, mean *WHO-5* 40.85±21.57), depressive symptoms (*WHO-5* 34.8% n = 108) and high work-related stress (PSS-10: 29.91±4.92). Senior leaders also had high levels of exhaustion (BAT: high/very high 89.0% n = 285) and psychosomatic complaints (experiencing muscle pain 48.2% n = 151) indicating burnout. Senior leaders in this study were engaging in self-endangering working behaviours (74.7% gave up leisure activities in favour of work, 63.4% sacrificed sufficient sleep), these were significantly higher in female senior leaders (work extensification males: 4.21 ± 0.62, females: 4.41 ± 0.50).

### Demographic characteristics

Tables 1 and 2 present the demographic characteristics of study participants by whole sample, country (Table 1) and gender (Table 2). In total, 323 school leaders participated in the *COVID-19 School Principals Study*, of which 172 were from Wales (53.25%) and 151 from Northern Ireland (46.75%) (Table 1). Of the whole sample, 67.5% (n = 216) were female. The sample consisted of 83.2% (n = 253) primary schools, and 80.6% headteachers (n = 257). School-level demographic characteristics reported by participants show the mean and median number of

**Table 1. Demographic characteristics (whole sample, n = 323, and by country).**

| | | % (n) mean ± standard deviation | | |
| --- | --- | --- | --- | --- |
| | | **Whole sample** | **Wales** | **Northern Ireland** |
| **Total** | | 323 | 172 (53.25%) | 151 (46.75%) |
| **Gender** | *Male* | 32.5% (104) | 63 (37.3%) | 41 (27.2%) |
| | *Female* | 67.5% (216) | 106 (62.7%) | 110 (72.8%) |
| | *Prefer not to say* | <5 | <5 | <5 |
| **School type** | *Primary* | 83.2% (253) | 130 (81.3%) | 123 (85.4%) |
| | *Secondary* | 16.8% (51) | 30 (18.7%) | 21 (14.6%) |
| **Leadership position** | *Headteacher* | 80.6% (257) | 133 (79.2%) | 124 (82.1%) |
| | *Deputy* | 8.2% (26) | 14 (8.3%) | 12 (7.9%) |
| | *headteacher Senior leadership* | 11.3% (36) | 21 (12.5%) | 15 (9.9%) |
| **Number of students in school** | *Mean* | 357.48 ± 333.10 | 396 ± 360.55 | 314.21 ± 294.82 |
| | *Median* | 240 | 250 | 220 |
| | *0–200* | 128 (40.1%) | 57 (33.9%) | 71 (47.0%) |
| | *201–400* | 96 (30.1%) | 54 (32.1%) | 42 (27.8%) |
| | *401–600* | 48 (15.0%) | 29 (17.3%) | 19 (12.6%) |
| | *601+* | 47 (14.7%) | 28 (16.7%) | 19 (12.6%) |
| **Percentage of students in school from socioeconomic groups** | *Low* | 43.17 ± 29.62 | 44.28 ± 31.06 | 42.08 ± 28.20 |
| | *Medium* | 46.19 ± 27.23 | 46.64 ± 28.55 | 45.75 ± 25.98 |
| | *High* | 7.70 ± 10.59 | 7.56 ± 10.88 | 7.84 ± 10.32 |
| **Percentage of pupils in school eligible for free school meals** | *0–20%* | 140 (43.8%) | 84 (48.8%) | 56 (37.1%) |
| | *21–40%* | 116 (36.3%) | 67 (39.0%) | 49 (32.5%) |
| | *41–60%* | 40 (12.5%) | 15 (8.7%) | 25 (16.6%) |
| | *61–80%* | 20 (6.3%) | <5 | 17 (11.3%) |
| | *81–100%* | 4 (1.3%) | 0 | <5 |

**Table 2. Demographic characteristics (by gender).**

| | | % (n) mean ± standard deviation | |
|---|---|---|---|
| | | **Male** | **Female** |
| **Total** | | 104 (32.2%) | 216 (66.9%) |
| **School type** | *Primary* | 70 (71.4%) | 182 (88.8%) |
| | *Secondary* | 28 (28.6%) | 23 (11.2%) |
| **Leadership position** | *Headteacher* | 81 (79.4%) | 175 (81.0%) |
| | *Deputy headteacher* | 7 (6.9%) | 19 (8.8%) |
| | *Senior leadership* | 14 (13.7%) | 22 (10.2%) |
| **Number of students in school** | *Mean* | 449.47 ± 397.49 | 309.54 ± 288.28 |
| | *Median* | 340 | 220 |
| | *0–200* | 32 (31.4%) | 96 (44.4%) |
| | *201–400* | 30 (29.4%) | 66 (30.6%) |
| | *401–600* | 14 (13.7%) | 33 (15.3%) |
| | *601+* | 26 (25.5%) | 21 (9.7%) |
| **Percentage of students in school from socioeconomic groups** | *Low* | 37.70 ± 26.49 | 45.77 ± 30.81 |
| | *Medium* | 49.71 ± 24.66 | 44.47 ± 28.37 |
| | *High* | 8.84 ± 10.80 | 7.19 ± 10.49 |
| **Percentage of pupils in school eligible for free school meals** | *0–20%* | 47 (45.6%) | 92 (42.6%) |
| | *21–40%* | 41 (39.8%) | 75 (34.7%) |
| | *41–60%* | 11 (10.7%) | 29 13.4%) |
| | *61–80%* | 4 (3.9%) | 16 (7.4%) |
| | *81–100%* | 0 | <5 |

students at senior leaders' schools (357.48 ± 333.10, 240). The percentage of pupils in low, medium and high socioeconomic groups within senior leaders' schools were reported (whole group; 43.17 ± 29.62, 46.19 ± 27.23, 7.70 ± 10.59). The majority of senior leaders worked within schools with 0–20% of pupils eligible for free school meals (43.8%).

Tables 3 and 4 present the descriptive statistics of senior leaders' working situation, well-being and work-related stress by whole sample, country (Table 3) and gender (Table 4). The distribution of responses can be viewed in the S4 Fig. Complete cases are presented, and responses with a minimum of five participants' data are presented for purposes of anonymity.

## Working situation

The mean weekly workload for senior leaders in this sample was 54.22 ± 11.3 hours (Wales: 54.93 ± 11.33, Northern Ireland: 53.43 ± 11.26), 76.8% (n = 241) reported working at least 50 hours per week during COVID-19. Weekly teaching responsibility was also reported as 10.29 ± 14.97 hours, differences between countries were observed (Wales: 7.12 ± 13.14, Northern Ireland 13.85 ± 16.09 hours). Workload increased during the COVID-19 pandemic compared to before, this was reported by 67.2% (n = 211) of the sample (Wales: 55.2%, n = 90; Northern Ireland: 80.1%, n = 121).

## General health

Senior leaders reported their perceived general health status, with 69.9% (n = 226) responding that this was good/very good (Wales: 79.4%, n = 135; Northern Ireland: 60.3%, n = 91). Male and female senior school leaders reported similar perceived health status, with 69.2% (n = 72) and 70.8 (n = 153) reporting this as good/very good, respectively.

**Table 3. Descriptive statistics of senior leaders' well-being, working situation and work-related stress (whole sample, country).**

| | | % (n) Mean ± standard deviation | | |
| --- | --- | --- | --- | --- |
| | | Whole sample | Wales | Northern Ireland |
| **Working situation** | *Teaching hours* | 10.29 ± 14.97 | 7.12 ± 13.14 | 13.85 ± 16.09 |
| | *Workload hours* | 54.22 ± 11.30 | 54.93 ± 11.33 | 53.43 ± 11.26 |
| **Workload compared to before COVID-19** | *Lower* | 12 (3.8%) | 8 (4.9%) | <5 |
| | *About the same* | 91 (29.0%) | 65 (39.9%) | 26 (17.2%) |
| | *Higher* | 211 (67.2%) | 90 (55.2%) | 121 (80.1%) |
| **General health** | *Good/very good* | 226 (69.9%) | 135 (79.4%) | 91 (60.3%) |
| | *Mean* | 2.09 ± 0.86 | 1.91 ± 0.81 | 2.27 ± 0.87 |
| **WHO-5** | *Mean* | 40.85 ± 21.57 | 44.93 ± 21.62 | 36.08 ± 20.59 |
| | *Low well-being* | 202 (65.2%) | 91 (54.5%) | 111 (77.6%) |
| | *Depressive symptoms* | 108 (34.8%) | 47 (28.1%) | 61 (42.7%) |
| **Sense of coherence** | *Mean* | 3.95 ± 1.21 | 4.16 ± 1.25 | 3.71 ± 1.13 |
| | *Comprehensibility* | 3.61 ± 1.26 | 3.83 ± 1.30 | 3.36 ± 1.16 |
| | *Manageability* | 3.33 ± 1.38 | 3.54 ± 1.42 | 3.10 ± 1.31 |
| | *Meaningfulness* | 4.83 ± 1.48 | 5.01 ± 1.49 | 4.58 ± 1.43 |
| **Exhaustion** | *Mean* | 3.74 ± 0.76 | 3.52 ± 0.80 | 4.00 ± 0.62 |
| | *High/very high* | 285 (89.0%) | 138 (81.2%) | 147 (98.0%) |
| **Psychosomatic complaints** | *Mean* | 2.70 ± 0.83 | 2.58 ± 0.85 | 2.83 ± 0.80 |
| **Self-endangering behaviour** | *Work extensification* | 4.35 ± 0.55 | 4.33 ± 0.51 | 4.36 ± 0.59 |
| | *Work intensification* | 4.20 ± 0.79 | 4.01 ± 0.84 | 4.42 ± 0.67 |
| | *Quality reduction* | 3.47 ± 0.76 | 3.31 ± 0.75 | 3.65 ± 0.73 |
| **Perceived stress** | *Total score* | 29.91 ± 4.92 | 28.72 ± 6.00 | 31.30 ± 2.65 |
| | *Helplessness* | 19.23 ± 4.14 | 18.32 ± 4.95 | 20.28 ± 2.57 |
| | *Self-efficacy* | 10.67 ± 1.94 | 10.37 ± 2.12 | 11.01 ± 1.66 |

## Well-being

Using the *WHO-5*, senior leaders' mean subjective psychological well-being was 40.85 ± 21.57 (Wales: 44.93 ± 21.62, Northern Ireland: 36.09 ± 20.59). 62.5% (n = 202) (Wales: 54.5%, Northern Ireland: 77.6%) were categorised as having low well-being using the cut-off score of ≤50 determined by Topp et al. [51] and 34.8% (n = 108) (Wales: 28.1%, n = 47, Northern Ireland: 42.7%, n = 61) were below the cut-off for depressive symptoms [52]. Male senior leaders reported higher subjective psychological well-being (42.9 ± 22.9) than female senior leaders (39.7 ± 20.90). Whilst there were minimal gender differences in being classified as having low well-being (males: 63.7%, n = 65 females: 66.2%, n = 137), females reported higher depressive symptoms (37.2%, n = 77) than males (30.4%, n = 31).

## Sense of Coherence

Findings regarding work-related SoC indicate that meaningfulness (the extent to which a work situation is seen as worthy of commitment and involvement) was rated the highest (4.83 ± 1.48, Wales: 5.01 ± 1.49, Northern Ireland: 4.58 ± 1.43), followed by comprehensibility (3.61 ± 1.26, Wales: 3.83 ± 1.30, Northern Ireland: 3.36 ± 1.16), with manageability rated lowest of the three sub-scales (3.33 ± 1.38, Wales: 3.54 ± 1.42, Northern Ireland: 3.10 ± 1.31). High mean values indicate stronger SoC, which can suggest coping more efficiently with work stressors (overall SoC whole sample: 3.95 ± 1.21, Wales: 4.16 ± 1.25, Northern Ireland: 3.71 ± 1.13). This trend was also observed between gender.

**Table 4. Descriptive statistics of senior leaders' well-being, working situation and work-related stress (gender).**

| | | % (n) mean ± standard deviation | |
| --- | --- | --- | --- |
| | | **Male** | **Female** |
| **Total** | | 104 (32.2%) | 216 (66.9%) |
| **Working situation** | *Teaching hours* | 9.54 ± 14.33 | 10.69 ± 15.29 |
| | *Workload hours* | 55.39 ± 10.87 | 53.63 ± 11.50 |
| **Workload compared to before COVID-19** | *Lower* | 5 (5.1%) | 7 (3.3%) |
| | *About the same* | 30 (30.3%) | 61 (28.5%) |
| | *Higher* | 64 (64.6%) | 146 (68.2%) |
| **General health** | *Good/very good* | 72 (69.2%) | 153 (70.8%) |
| | *Mean* | 2.07 ± 0.98 | 2.09 ± 0.84 |
| **WHO-5** | *Mean* | 42.9 ± 22.91 | 39.79 ± 20.90 |
| | *Low well-being* | 65 (63.7%) | 137 (66.2%) |
| | *Depressive symptoms* | 31 (30.4%) | 77 (37.2%) |
| **Sense of coherence** | *Mean* | 3.90 ± 1.25 | 3.99 ± 1.18 |
| | *Comprehensibility* | 3.58 ± 1.34 | 3.64 ± 1.21 |
| | *Manageability* | 3.33 ± 1.39 | 3.34 ± 1.37 |
| | *Meaningfulness* | 4.69 ± 1.50 | 4.91 ± 1.45 |
| **Exhaustion** | *Mean* | 3.60 ± 0.83* | 3.81 ± 0.71* |
| | *High/very high* | 87 (83.7%) | 197 (91.6%) |
| **Psychosomatic complaints** | *Mean* | 2.39 ± 0.81* | 2.84 ± 0.80* |
| **Self-endangering behaviour** | *Work extensification* | 4.21 ± 0.62* | 4.41 ± 0.50* |
| | *Work intensification* | 4.09 ± 0.84 | 4.25 ± 0.76 |
| | *Quality reduction* | 3.43 ± 0.81 | 3.48 ± 0.73 |
| **Perceived stress** | *Mean* | 29.26 ± 5.56 | 30.21 ± 4.56 |
| | *Helplessness* | 18.56 ± 4.68 | 19.52 ± 3.79 |
| | *Self-efficacy* | 10.71 ± 2.06 | 10.66 ± 1.89 |

* Depicts statistically significant differences between groups

### Exhaustion and psychosomatic complaints

Regarding exhaustion, mean values for the sample of senior leaders were 3.74 ± 0.76. (Wales: 3.52 ± 0.80, Northern Ireland 4.00 ± 0.62), and females (3.81 ± 0.71) reported significantly higher exhaustion compared to males (3.60 ± 0.83, t = -2.208, p = 0.029). Applying the statistical norms proposed by Schaufeli, De Witte and Desart [54] suggest that 89.0% (n = 285) of senior leaders exhibit high/very high exhaustion (Wales: 81.2%, n = 183, Northern Ireland: 98%, n = 147), and gender differences (males: 83.7%, n = 87, females: 91.6%, n = 197).

Further descriptive data (S4 Fig) shows that senior leaders from Wales and Northern Ireland report often/always feeling mentally exhausted (57.1% and 82%, respectively) and physically exhausted (37.1% and 77.7%, respectively). Exhaustion is a core symptom of work-related burnout, with secondary symptoms including psychosomatic complaints, that is experiencing physical symptoms that are attributed to or exacerbated by mental stress. Most notably, this included often/always experiencing muscle pain such as neck, shoulder and back (Wales: 39.5%, Northern Ireland: 58.2%) and headaches (Wales: 31.9%, Northern Ireland: 38.7%. Statistically significant gender differences were also observed, with female senior leaders reporting higher psychosomatic complaints (females: 2.84 ± 0.80, high/very: 41.2%, males: 2.39 ± 0.81, high/very high: 16.1%, t = -4.721, p = <0.001).

### Self-endangering behaviour

The self-endangering work behaviour scale [55] assessed three-sub scales; work extensification, work intensification and quality reduction was used. Regarding extensification of work (i.e. extending working hours), mean values for the senior leaders in this sample were 4.35 ± 0.55. The distribution of responses indicates that 91.5% (n = 293) reported to fairly often/very often worked extra hours in the previous three months, 74.7% (n = 239) given up leisure activities in favour of work, and 63.4% (n = 202) sacrificing sufficient sleep. Differences between genders were statistically significant (t = -2.879, p = 0.005), with females reporting higher extensification of work than their male counterparts, 4.41 ± 0.50 and 4.21 ± 0.62, respectively. For work intensification, that is, working at an increased pace and multitasking whilst limiting break periods and work-related social interactions [58], mean values were 4.20 ± 0.79 (whole sample). The majority of senior leaders (fairly often/very often: 74.4%, n = 239) reported to work at a pace they found burdensome, that cannot be sustained in the long term (81.3%, n = 261), and they know it is not good for them (82.9%, n = 266). The third sub scale, quality reduction, reducing the quality of work in reaction to excessive work demands, was rated the lowest by senior leaders (mean values; 3.47 ± 0.76).

### Perceived stress

Using the 10-item *Perceived Stress Scale* adapted to the COVID-19 working context for the purpose of this study, findings from the current study present total scores for the sample of senior leaders (29.91 ± 4.92), and sub-scales of perceived helplessness (19.23 ± 4.14), and perceived self-efficacy (10.67 ± 1.94. Gender differences suggest significantly greater perceived helplessness reported by females than males (19.52 ± 3.79 and 18.56 ± 4.68, respectively, t = -1.996, p = 0.047). Within items from this sub-scale, nearly half of females (49.6%) reported that they could not cope with all their work tasks (fairly often/very often) compared to 28.8% of males.

## Discussion

This study aimed to examine the working situation, well-being and work-related stress of senior school leaders in Wales and Northern Ireland and explore gender differences during a period of educational crisis leadership within the COVID-19 pandemic between 2021 and 2022. This is part of an ongoing international study through the COVID-HL network, with specific relevance to current and emerging policy and practice in Wales and Northern Ireland. Findings suggest that senior school leaders reported high workloads, low well-being, depressive symptoms and high work-related stress. Senior leaders also had high levels of exhaustion and psychosomatic complaints indicating burnout and were engaging in self-endangering working behaviours, these were significantly higher in female senior leaders.

During the COVID-19 pandemic, senior school leaders were exposed to a period of "crisis leadership" [10]. Whilst evidence has focussed on developing frameworks to better understand how school leaders can manage and navigate periods of crisis leadership, there is a dearth of evidence examining the impacts this can have and how they experience this unprecedented disruption. Findings from this study contribute to the limited knowledge base understanding the impact this period of crisis leadership had on specific aspects of school leaders' physical and mental health. This is important in order to enable the education and health systems to respond with a targeted approach and appropriate resources, and in addressing the *recovery* and *learning* phases of educational crisis leadership models based on the experience of school leaders [13]. This evidence base can shape and inform policy and practice and directly address the working situation and well-being experiences of leaders in the education system. This

should be considered fundamental components embedded within frameworks of educational crisis leadership and future crises. The findings in the current study supports other work in this area both nationally and internationally [20, 33–37, 59].

The majority of senior leaders in this study reported low well-being, lower than the general adult population pre-pandemic [60] and COVID-HL international study data from Hong Kong [33, 34] and Switzerland [35]. Elsewhere, reports from 2020 to 2022 suggest negative impacts of the sustained work-related pressures on senior leaders' well-being and mental health [20, 61, 62]. In this study, females exhibited higher depressive symptoms than males, reflecting global data that suggests existing gender disparities in the prevalence of depression widened further during COVID-19 [63]. Findings regarding the well-being and mental health of senior school leaders in this study have wider implications as evidence demonstrates associations between the well-being of educational practitioners and pupil-level outcomes, including health, well-being and educational attainment [64].

The large majority of senior leaders in this study reported a higher workload than before the pandemic and working at least 50 hours per week. UK-based educational practitioners work longer hours than their international colleagues [65], and headteachers in the UK reported to work 20 hours more per week during COVID-19 than classroom teachers [66]. These findings are concerning given that high workload has been identified as a driver in school leadership attrition and a key reason for staff considering leaving the profession [23, 61], headteacher and teacher retention rates were a concern pre-pandemic and this has since been labelled a "crisis" in England, Wales and Northern Ireland [16, 67]. Available data for Wales shows an increase in secondary senior leaders leaving the profession over the last 10 years, with the largest number leaving in 2022 [24]. Whilst this data is only available for senior leaders in secondary schools, there has also been a steady increase in all teaching staff leaving the profession (e.g. n = 414 in 2020, n = 756 in 2022) [24]. Further, a survey of over 2,000 senior leaders across England, Wales and Northern Ireland suggests increasing dissatisfaction of school leadership as a career choice and fewer aspirations of those in deputy/assistant roles to progress to senior headship level [23]. Most recently, senior leaders in Wales and Northern Ireland have resorted to industrial action, citing workload, below-inflation pay awards and chronic school underfunding as key concerns within the profession [68–70]. This suggests these work-related pressures persist, and industrial action is ongoing at the time of writing [71].

In this study, senior leaders reported often giving up leisure activities in favour of work, sacrificing sufficient sleep and waiving breaks during the working day. This relates to the transactional stress model, which explains individuals' cognitive and behavioural responses as coping mechanisms to manage internal or external perceived stressors [72]. Females were significantly more likely to report self-endangering behaviours, in agreement with research in Germany which found that females were more likely to show overcommitting behaviours within their role, overexerting themselves through excessive work engagement [73]. Furthermore, findings from the teaching profession in Switzerland reported that prolonging working hours mediates the relationship between work overload and exhaustion, thus, work extensification is used as a coping mechanism to high working demands [74].

The majority of senior leaders reported that this level of work was burdensome, cannot be sustained in the long Term and was not good for them. Whilst our findings are in line with international study data from Hong Kong, [34], the prevalence of self-endangering behaviours was higher in Wales and Northern Ireland. Given that high work-related pressures are still being reported in the UK [67], it is reasonable to assume that many senior leaders are still engaging in self-endangering behaviours as a mechanism to mediate the negative work-related stressors associated with continued high workloads and working conditions. Whilst necessary to fulfil working demands, this is not conducive to their health and well-being.

Increased workload and self-endangering behaviours can lead to burnout, a psychological syndrome caused by chronic job stressors [75], of which exhaustion is a core symptom [54]. Senior leaders in this study reported very high levels of exhaustion, this is supported by research nationally during COVID-19 which demonstrates that headteachers and senior leaders reported the highest exhaustion compared to other educational practitioners [59]. Exhaustion was also higher in the current study than in international COVID-HL data [37]. Our study also found statistically significant gender differences, with females exhibiting higher exhaustion. These gender differences in school leader populations have also been noted in Sweden [26]. In a review of the role of gender in workplace stress, Gyllensten and Palmer identified a number of contributing factors for females, including increased responsibility for domestic chores, work-family conflicts and additional caring responsibilities (e.g. childcare and elderly parents) [76]. Though this review is broad and considers multiple occupations and not solely senior school leaders, it offers insights into gender differences observed in this study and reflects wider societal gender expectations and norms.

Psychosomatic complaints, recognised as physical symptoms attributed to or exacerbated by mental stress, are considered secondary symptoms of burnout [54]. Senior leaders in this study also reported higher psychosomatic complaints than reported internationally [33], including higher frequencies of headaches and muscle pain (e.g. neck, shoulder, back), feeling mentally and physically exhausted and finding it difficult to recover their energy after a day of work [37]. Again, significant gender differences were observed, with females reporting higher psychosomatic complaints. Senior leaders in this study may have been at risk of, or were experiencing burnout during the COVID-19 pandemic, as reported elsewhere in the UK during this time [59].

Further to this and building on the transactional stress model are findings relating to senior leaders' perceived stress, this is minimally lower in Wales and Northern Ireland than reported in Hong Kong [33], though significantly higher than those in Taiwan [36]. Outside of the COVID-HL international study, findings from Wales and Northern Ireland are also significantly higher than teachers elsewhere in the UK [77], high levels of perceived stress amongst school leaders have also been found elsewhere in Wales [22]. Higher perceived stress has been associated with other outcomes, including hormonal changes and disturbances to the menstrual cycle in women [78], with females in this study reporting higher perceived helplessness, insomnia [79] and accessing primary care services [80]. A framework that explains how people manage stressful situations to maintain their health and well-being, a *Sense of Coherence* (SoC) and work-related SoC are used to indicate the health-promoting quality of life at work [53]. Senior leaders in this study reported manageability as the lowest of the three sub-scales, suggesting they perceived inadequate resources available to cope with the demands of the pandemic. According to the job demands-resources model, job resources (physical, psychological, social or organizational) are a central component in achieving work goals, reducing job demands and negating the associated physiological/psychological impacts [81]. Within this theoretical model, it is proposed that adequate job resources can buffer the impact of job demands on job-related strains such as burnout. This emphasizes the importance of necessary personal, organisational and governmental support for senior leaders and could be one consideration in reducing levels of burnout.

Despite the numerous work-related challenges experienced by senior leaders in this study, meaningfulness, the extent to which a person perceives their work situation as worthy of commitment and involvement, was reported the highest. Senior leaders still valued their role and its contribution, this is supported elsewhere within qualitative work with educational practitioners remaining in challenging working situations who expressed a strong sense of social responsibility within their role [82]. Following a similar trend to other measures in this study,

overall SoC was lower than comparable international study data [34, 36]. Higher work-related SoC predicted higher subjective well-being scores in Hong Kong [34], and was associated with lower perceived stress and lower likelihood of depressive symptoms in Taiwan [36]. Thus, this indicates stronger SoC acting as a protective factor in the stress, well-being and mental health of senior school leaders, and factors to foster SoC, including adequate resources must be considered.

Senior leaders in this study reported high general health status, contradicting wider study findings regarding low well-being and high work-related stress. How the term 'general health' was interpreted by senior leaders is unclear, this could be perceived as encompassing physical and/or psychological dimensions. It is plausible that whilst leaders in this study report work-related stress which impacts well-being, they may engage in health-promoting behaviours (e.g. physical activity), and thus perceive their health status as high. This data were not captured and requires further study, amongst a paucity of evidence regarding the presence of health-promoting behaviours in this population. This high general health status reported by senior leaders in this study may also be explained through SoC, with unpublished evidence from Sweden indicating the protective influence of SoC in reducing the impact of work-related stress on general health [83]. This is further supported by longitudinal research in the UK that suggests a salutogenic effect of strong SoC on chronic disease incidence [84]. Whilst associations between variables were not explored in this study, this warrants the need for further research to examine associations between perceived general health with other well-being and work-related indicators. Furthermore, although senior leaders in this study report lower well-being than the general population, perceived general health status is comparable to Census data collected in England and Wales in 2021 [85]. This is in line with research from France demonstrating comparable general health status between educational practitioners and other professions [86]. Senior leadership is considered a high occupational grade, this may offer protection for physical health as evidenced by findings in the Whitehall and Whitehall II cohort studies regarding occupation rank and psychosocial work environment [87, 88].

Ultimately, these findings regarding senior leaders' work-related stress and well-being are concerning, given that Wales and Northern Ireland are currently experiencing a "crisis" in educational leadership against a backdrop of pandemic-related pressures. Assumptions regarding 'bouncing back' have been short-lived; more recent evidence suggests these initial pandemic-related pressures have accumulated with prior strain experienced by senior leaders and are indeed long-lasting. Survey research from the most recent *Headteacher Wellbeing Index* in the UK indicates that the stress levels of educational practitioners increased up to 2022 [20], and evidence examining post-pandemic well-being and burnout from other fields such as healthcare highlights this further [89]. International evidence has demonstrated the importance of organisational preparedness, ensuring capacities are in place across systems and settings to safeguard the well-being of those at the forefront of decision making and leadership [90]. This should be considered in times of crisis and periods of higher work-related demands requiring adaptations to roles and responsibilities [91].

The high attrition rates of senior leaders and other educational practitioners, and an increasing dissatisfaction of progression to leadership positions further exacerbate this [23]. This proves costly to educational systems and places additional financial and other pressure on educational settings and policy response. This not only has implications for senior leaders' well-being, work-related stress and issues regarding retention and leadership progression within the profession but also on pupil-level outcomes, including health, well-being and educational attainment [30, 64, 92].

This study expands current knowledge on the impacts of the COVID-19 pandemic and "*crisis leadership*" on educational leaders in Wales and Northern Ireland. Applying the findings

through the lens of educational crisis leadership frameworks, it is essential to consider the last phases of *recovery* and *learning* to minimise and address impacts and increase preparedness [13]. Findings suggest that both tailored and targeted support from the education and health sectors and government are urgently required to improve and optimise the working conditions of senior leaders. Based on the findings from this study, we propose five recommendations for consideration across research, policy and practice based on lessons learnt and what may mitigate these issues in any future national crises:

i. A more strategic approach to supporting the well-being of educational leaders in Wales and Northern Ireland is essential, including joint working between the health/social care and education sectors to provide mental health support to senior leaders.

ii. In the short term, the mental health and well-being supports currently available from both the education and health/social care sectors should be more explicitly highlighted to senior school leaders.

iii. Further research into what mental health support and resources are needed at individual, organisational and systems level to better support senior leaders in their role.

iv. Greater clarity on the extent and quality of leadership development provision to specifically support leaders' well-being is needed. This is particularly important during periods of major education system-level reforms; leadership is critical to support school and system improvement and ensuring ownership by practitioners.

v. Further research charting changes over time in leaders' experience of their well-being can contribute to strengthening the evidence base in this area. This includes longitudinal research using both quantitative and qualitative methods, extending this research to include senior leaders from nursery and post-16 educational settings and capturing this across the four nations of the UK to contribute to international data.

## Strengths and limitations

This study builds on the evidence base of the impacts of the COVID-19 pandemic on educational settings and extends this to senior school leaders during a period of "*crisis leadership*" in Wales and Northern Ireland. This is part of a wider global study through the COVID-HL network, enabling international comparisons. There are limitations to consider when interpreting the findings from this study. Whilst efforts were made to invite a range of participants through publicly available school contact information and key education stakeholders to increase the response rate, this study reports results obtained from a convenience sample of participants that chose to complete the survey and may not represent the wider senior leadership population. Whilst this study extends our understanding of the impact on school leaders, other educational practitioners are likely to have experienced similar work-related pressures and capturing a wider range of roles and experiences would provide further context to school settings, further research is required here. There is potential for response bias, and the reporting of socially desirable responses. In addition, given the high levels of stress and exhaustion reported in this study, selection bias is possible, the sample consists of more females, and it may be possible that more stressed or exhausted female senior leaders may have chosen to participate in the survey. The cross-sectional self-report survey was administered to a convenience sample of senior leaders in Wales and Northern Ireland at different time points during the COVID-19 pandemic, thus, it would not be helpful to compare between countries. Both countries implemented various public measures at different timepoints, including the closure and

reopening of face-to-face education settings. Further longitudinal research using both quantitative and qualitative methods is required, obtaining a representative sample and extending this research across the four nations of the United Kingdom.

## Conclusions

Findings in this study suggest that senior school leaders in Wales and Northern Ireland reported high workloads, low well-being, depressive symptoms, and high work-related stress during a period of crisis leadership in the COVID-19 pandemic. Senior leaders also had high levels of exhaustion and psychosomatic complaints indicating burnout and were engaging in self-endangering working behaviours such as sacrificing sufficient sleep. Gender differences were observed, with females reporting statistically higher exhaustion, psychosomatic complaints, and extensification of work (e.g. extending working hours) than their male counterparts. These findings are concerning, not least because concerns were raised prior to the pandemic regarding the well-being, mental health and working conditions of senior school leaders across the UK, with experts warning of a potential crisis in leadership in education due to the high numbers leaving the profession [4, 11, 18, 23]. This has also been observed across other education sectors in the UK, especially higher education [5]. It is likely the COVID-19 pandemic exacerbated this, and further research is required to examine the long-term impacts at individual and organisational-level and within other educational settings and contexts. It is vital to understand the impact the COVID-19 pandemic had on specific aspects of school leaders' physical and mental health to enable the education and health systems to respond with a targeted approach and appropriate resources. This is especially relevant with all four nations of the UK having undergone, or are currently undergoing, major education system-level reforms [93], including new national curricula and qualifications. For example, with the start of the new Curriculum for Wales from September 2022 onwards; leadership is critical to support school and system improvement, as well as ensuring ownership by practitioners [94]. This evidence base can thus shape and inform emerging policy and practice to directly address the working situation and well-being experiences of leaders in the education system.

## Supporting information

**S1 Checklist. PLOS inclusivity in global research questionnaire.**
(DOCX)

**S1 Fig.** *COVID-HL School Principals Survey* **question categories and sub-categories.**
(DOCX)

**S2 Fig. Wales Survey: Full** *COVID-19 School Principals Survey* **Wales.**
(PDF)

**S3 Fig. Northern Ireland Survey: Full** *COVID-19 School Principals Survey* **Northern Ireland.**
(PDF)

**S4 Fig.** *COVID-HL School Principal Survey* **distribution of responses.**
(DOCX)

## Acknowledgments

The authors would like to thank the senior school leaders who gave up their time to participate in this research study, particularly during such a challenging period within their roles, we are

grateful for their participation. We would like to thank the National Academy for Educational Leadership Wales for their support with this study. Finally, we would like to thank Professor Kevin Dadaczynski and Professor Orkan Okan for their invitation to join the COVID-HL network and their support and guidance with undertaking this research in Wales and Northern Ireland.

## Author Contributions

**Conceptualization:** Emily Marchant, Joanna Dowd, Lucy Bray, Gill Rowlands, Tom Crick, Kevin Dadaczynski, Orkan Okan.

**Data curation:** Emily Marchant, Joanna Dowd, Nia Miles, Michaela James.

**Formal analysis:** Emily Marchant.

**Funding acquisition:** Emily Marchant, Tom Crick.

**Investigation:** Emily Marchant, Joanna Dowd, Lucy Bray, Tom Crick, Kevin Dadaczynski, Orkan Okan.

**Methodology:** Emily Marchant, Gill Rowlands, Kevin Dadaczynski, Orkan Okan.

**Project administration:** Emily Marchant, Joanna Dowd, Michaela James.

**Resources:** Emily Marchant, Lucy Bray, Gill Rowlands, Nia Miles, Kevin Dadaczynski, Orkan Okan.

**Software:** Emily Marchant.

**Supervision:** Lucy Bray, Gill Rowlands, Tom Crick, Kevin Dadaczynski, Orkan Okan.

**Writing – original draft:** Emily Marchant.

**Writing – review & editing:** Emily Marchant, Joanna Dowd, Lucy Bray, Gill Rowlands, Nia Miles, Tom Crick, Michaela James, Kevin Dadaczynski, Orkan Okan.

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
