## [Decision Letter · Decision Letter 0]

9 Nov 2023

PONE-D-23-26570The well-being and work-related stress of senior school leaders in Wales and Northern Ireland during the COVID-19 pandemic: A cross-sectional descriptive studyPLOS ONE

Dear Dr. Marchant,

Thank you for submitting your manuscript to PLOS ONE. After careful consideration, we feel that it has merit but does not fully meet PLOS ONE’s publication criteria as it currently stands. Therefore, we invite you to submit a revised version of the manuscript that addresses the points raised during the review process.

We look forward to receiving your revised manuscript.

Kind regards,

Beatriz Talavera-Velasco

Academic Editor

PLOS ONE

Journal Requirements:

3. Please include a complete copy of PLOS’ questionnaire on inclusivity in global research in your revised manuscript. Our policy for research in this area aims to improve transparency in the reporting of research performed outside of researchers’ own country or community. The policy applies to researchers who have travelled to a different country to conduct research, research with Indigenous populations or their lands, and research on cultural artefacts. The questionnaire can also be requested at the journal’s discretion for any other submissions, even if these conditions are not met.  Please find more information on the policy and a link to download a blank copy of the questionnaire here: https://journals.plos.org/plosone/s/best-practices-in-research-reporting. Please upload a completed version of your questionnaire as Supporting Information when you resubmit your manuscript.

5. We note that S1 Fig and S2 Fig in your submission contain copyrighted images. All PLOS content is published under the Creative Commons Attribution License (CC BY 4.0), which means that the manuscript, images, and Supporting Information files will be freely available online, and any third party is permitted to access, download, copy, distribute, and use these materials in any way, even commercially, with proper attribution. For more information, see our copyright guidelines: http://journals.plos.org/plosone/s/licenses-and-copyright.

1. You may seek permission from the original copyright holder of S1 Fig and S2 Fig to publish the content specifically under the CC BY 4.0 license.

Reviewers' comments:

Reviewer's Responses to Questions

**Comments to the Author**

1. Is the manuscript technically sound, and do the data support the conclusions?

Reviewer #1: Yes

2. Has the statistical analysis been performed appropriately and rigorously? 

Reviewer #1: Yes

3. Have the authors made all data underlying the findings in their manuscript fully available?

Reviewer #1: Yes

4. Is the manuscript presented in an intelligible fashion and written in standard English?

Reviewer #1: Yes

5. Review Comments to the Author

Reviewer #1: Overall, I thought this was a well-written paper. Most of my feedback is rather minor. The major issue I believe worth highlighting is the need for some sort of conceptual/theoretical framing section. I think rooting this study within crises, leadership, and what research has come out about COVID-19 in education is needed (presently, not enough has been inserted). Otherwise, most of the paper is strong and the claims are supported. Please see comments below.

Intro

Page 9, line 25: this sentence read a little awkward. I recommend revising and shortening.

Page 9, line 29: I wouldn’t say it impacted everyone, but rather most people. Some contexts didn’t change a thing.

Page 11, line 74: how many people were in the convenience sample, for both Whales and Northern Ireland.

General: After the intro, some sort of conceptual framing or theory section is needed to illuminate your findings. Something around crisis leadership and management would be useful. Also, expand on what current COVID-19 research in education has found. This is a key piece missing right now. Tie these back to the various concepts the survey studied. Theory helps to bring clarity.

Methods

General: I think a paragraph about the contexts of Wales and Northern Ireland in response to the pandemic is needed. This is your context, so help the reader understand it a little more.

Page 12: say how many questions the survey had in the actual text, and give some key examples. Help the reader better understand its scope in the text. Also, there are various categories in the survey you discuss. First, upfront, say each of the categories that you use (as a list), then expand upon them later (as you do). This helps the reader follow your organization of information. Also, about how long did it take to complete the full survey? Again, just give a little more detail upfront. More is better here.

Data analysis: This section seemed far too short. Give a little more color to it. Two sentences is not enough.

Results

General: It seems like the health and well-being data contradicted each other (to a degree). Thus, some discussion of this is required in the discussion section. I would have assumed lower health given the other subsection scores.

General: Do you have more data on the retention rates of senior leaders in Wales and Northern Ireland? If so, that would reinforce the claims you make by a lot. Tying to actual burnout levels now would be great.

Page 16: Before breaking down each category, I recommend some broad, interesting findings in a short paragraph. Give the reader a taste, then expand upon those findings in each subsection (as you do). Remember, show the forest, then the trees! This is common practice.

6. PLOS authors have the option to publish the peer review history of their article (what does this mean?). If published, this will include your full peer review and any attached files.

Reviewer #1: **Yes: **Craig De Voto

---

## [Author Response · Author response to Decision Letter 0]

12 Dec 2023

Response to reviewer

Firstly, on behalf of the co-authors, we would like to express our sincere gratitude to the reviewer for their productive and positive comments regarding the manuscript. This review process has significantly improved the manuscript, and we are grateful for the time taken to provide such useful feedback. 

Introduction

Page 9, line 25: this sentence read a little awkward. I recommend revising and shortening. 

Thank you for drawing our attention to this, we have reworded the first five lines of the introduction to improve clarity of expression: 

Page 3 line 24-29:

The COVID-19 pandemic caused by the transmission of severe acute respiratory syndrome coronavirus 2 (SARS-CoV-2) resulted in unprecedented societal changes. In response, a range of public health measures were implemented to reduce social contacts and transmission of SARS-CoV-2. This included changes to the delivery of learning, teaching and assessment within educational settings and contexts. Measures included full or partial closure of face-to-face provision, a move to hybrid and blended learning, and the introduction of a variety of school-based measures upon the full return to education [1–6].

Page 9, line 29: I wouldn’t say it impacted everyone, but rather most people. Some contexts didn’t change a thing. 

We have addressed this by rewording from all people, to all layers of education, thus, whilst not all of the education population were impacted, there were impacts at each layer from pupils through to leadership teams. 

Page 3 line 30:

These significant and prolonged changes to the delivery of teaching and learning impacted all layers of education.

Page 11, line 74: how many people were in the convenience sample, for both Whales and Northern Ireland. 

We have now updated the manuscript with sample information: 

Page 7 line 192-193:

A convenience sample of participants (total: n=323, Wales: n=172, Northern Ireland: n=151) were recruited via email, social media and through education stakeholders.

General: After the intro, some sort of conceptual framing or theory section is needed to illuminate your findings. Something around crisis leadership and management would be useful. Also, expand on what current COVID-19 research in education has found. This is a key piece missing right now. Tie these back to the various concepts the survey studied. Theory helps to bring clarity. 

Thank you for highlighting this important conceptual framework suggestion. We agree with your points and believe more direct framing of the study through the theoretical lens of educational crisis leadership would significantly strengthen the manuscript.

We have incorporated more explicit discussions of educational crisis leadership conceptual frameworks throughout the manuscript, including the Introduction, Methods and Materials, Discussion and Conclusion, please find below details of these revisions:

Title

Page 1:

The well-being and work-related stress of senior school leaders in Wales and Northern Ireland during COVID-19 “educational leadership crisis”: A cross-sectional descriptive study

Page 2 line 5-6:

leaders were exposed to high demands relating to the numerous challenges they had to manage during a “crisis leadership” period.

Page 3 line 40-42: 

Senior school leaders are responsible for all aspects of school life and therefore had to cope with high demands due to the COVID-19 pandemic, a period defined as crisis leadership [10,11].

Page 3-4 line 43-79:

The term crisis leadership is characterised as an urgent situation requiring immediate and decisive action by an organisation or leaders of the organisation [12]. In the context of education, evidence regarding crisis leadership is limited but explores how educational leaders manage a period of “unexpected, fundamental disruption to school functioning with potentially high consequences for the organization, its stakeholders, and its reputation” [13, p.315]. To date, scoping reviews demonstrate the majority of research focuses on the response to and recovery from natural disasters and human-made crises, with a significant lack of research on educational leadership during public health crises [14]. However, crisis leadership in education was brought to the forefront during COVID-19, and a growing body of evidence has drawn attention to its importance in securing stability for learners, parents, staff and the community during a time of unprecedented disruption [9,12,13,15]. Recent work has focussed on developing frameworks to understand the processes behind the management of educational crises, such as the five phase cycle of mitigation/prevention, preparedness, response, recovery and learning [13]. Whilst our understanding of the conditions that enable effective educational crisis leadership is growing [13,14], these frameworks focus on the skills and attributes of leaders during the process of crisis leadership, neglecting outcomes such as how school leaders deal with and are impacted by this period. Thus, there is a gap in empirical research examining the impact on how school leaders experienced crisis leadership during COVID-19. This is important because factors such as well-being and work-related stress may be impacted which are fundamental during and after periods of crisis leadership.

Page 5 line 141-143:

This study aimed to examine the working situation, well-being and work-related stress of senior school leaders in Wales and Northern Ireland and explore gender differences as a whole sample in a period of educational crisis leadership during the COVID-19 pandemic 2021 and 2022. 

Page 7 line 178-186:

Conceptual framework

This study is conceptually positioned within the framing of educational crisis leadership. Defined above, this study recognises this as a period of “unexpected, fundamental disruption to school functioning with potentially high consequences for the organization, its stakeholders, and its reputation” [13, p.315] that requires immediate and decisive action by school leaders [12]. Whilst this study does not aim to examine the conditions and factors associated with effective educational crisis leadership, it aims to examine how school leaders experienced crisis leadership and the perceived impact this had on their working situation, well-being and work-related stress. These are currently neglected areas of study with the literature. In recognition of current frameworks on educational crisis leadership [13], this study informs the recovery and learning phases of these models based on school leaders’ experiences.

Page 19 line 443-454:

During the COVID-19 pandemic, senior school leaders were exposed to a period of “crisis leadership” [10]. Whilst evidence has focussed on developing frameworks to better understand how school leaders can manage and navigate periods of crisis leadership, there is a dearth of evidence examining the impacts this can have and how they experience this unprecedented disruption. Findings from this study contribute to the limited knowledge base understanding the impact this period of crisis leadership had on specific aspects of school leaders’ physical and mental health. This is important in order to enable the education and health systems to respond with a targeted approach and appropriate resources, and in addressing the recovery and learning phases of educational crisis leadership models based on the experience of school leaders [13]. This evidence base can shape and inform policy and practice and directly address the working situation and well-being experiences of leaders in the education system. This should be considered fundamental components embedded within frameworks of educational crisis leadership and future crises. The findings in the current study supports other work in this area both nationally and internationally [20,33–37,59].

Page 24-25 line 587-597:

This study expands current knowledge on the impacts of the COVID-19 pandemic and crisis leadership on educational leaders in Wales and Northern Ireland. Applying the findings through the lens of educational crisis leadership frameworks, it is essential to consider the last phases of recovery and learning to minimise and address impacts and increase preparedness [13].

Methods

General: I think a paragraph about the contexts of Wales and Northern Ireland in response to the pandemic is needed. This is your context, so help the reader understand it a little more.

We agree that adding both general and Covid-related context to Wales and Northern Ireland would be useful for the reader. To address your suggestion, we have included an additional sub-section in this chapter: 

Page 6 line 153-168:

Context

Education and health are devolved responsibilities of the respective governments of Wales and Northern Ireland within the UK. There was some general consistency in both countries’ educational response to COVID-19 including periods of closure and the reopening of face-to-face educational provision, blended and hybrid learning, and the implementation of school-based measures (e.g. social distancing) [38,39]. However, there was variation in the timings of implementation and relaxation of measures during the entirety of the COVID-19 pandemic and within the study period [38,39].

In Wales, the collection of data shortly followed the re-opening of face-to-face education (February-April 2021), and encompassed requirements of self-isolation and hybrid learning (June and July 2021), teacher-based assessments (June 2021), increases in COVID-19 cases linked to schools (July and October 2021), regular lateral flow testing of pupils (secondary) and teachers (September 2021) and the reintroduction of school-based public health measures (wearing face coverings). In addition to this, schools across Wales were preparing to roll out a new curriculum, the Curriculum for Wales [40] following significant education reform [41]. In Northern Ireland, the data collection period was later into the pandemic. Whilst education had returned somewhat to pre-pandemic delivery, this was amongst a backdrop of continued high COVID-19 rates, significant staff and pupil absences (January and February 2022), and ongoing COVID-related impacts and sustained pressures on the education system [39]. 

We have also further addressed this as a limitation of the study:

Page 26 line 661-663:

Both countries implemented various public measures at different timepoints, including the closure and reopening of face-to-face education settings.

Page 12: say how many questions the survey had in the actual text, and give some key examples. Help the reader better understand its scope in the text. Also, there are various categories in the survey you discuss. First, upfront, say each of the categories that you use (as a list), then expand upon them later (as you do). This helps the reader follow your organization of information. Also, about how long did it take to complete the full survey? Again, just give a little more detail upfront. More is better here. 

To address this, we have provided information in-text about categories and sub-categories (see below) and an additional supplementary file (S1 Fig) that contains a full breakdown of all categories and sub-categories of the full survey. 

Page 8 line 213-226:

The COVID-19 School Principal Survey [32] was designed by the international COVID-HL network and contained 33 questions. The original survey consisted of various categories and sub-categories; socio-demographic information (e.g. gender, type of school), current work situation (e.g. Sense of Coherence, perceived stress), health information in the context of COVID-19 (e.g. health literacy), health promotion and prevention in school (e.g. health promoting school factors) and health situation (general health, well-being). A full breakdown of categories and sub-categories is presented in S1 Fig. For the purpose of this study, questions relating to work situation, health, well-being, work-related Sense of Coherence, exhaustion and psychosomatic complaints, self-endangering behaviour and perceived stress were used for analyses. The survey was adapted for Wales and Northern Ireland including country specific wording, additional questions seeking demographic characteristics, and minor re-structuring (S2 and S3 Fig). The inclusion criteria for the survey were any senior leadership staff (headteacher, deputy headteacher, senior leadership team). The survey was completed on Microsoft Forms and took an average completion time of 30 minutes for participants in Wales and 28 minutes for participants in Northern Ireland. A full copy of the survey for Wales and Northern Ireland is available in the S2 and S3 Fig. A full copy of the original survey is available upon request.

For the purpose of providing this contextual information for the reader we feel this is best placed in the supplementary information to avoid large word limits. We have also included an asterisk to denounce sub-categories of questions used within analyses. As categories and sub-categories, full surveys for Wales and Northern Ireland are available in the supplementary information, and individual question and response examples are provided in the Methods and materials section, we feel this sufficiently provides the information the reader requires. 

We have added in average completion time of the survey:

Page 8 line 223-224:

The survey was completed on Microsoft Forms and took an average completion time of 30 minutes for participants in Wales and 28 minutes for participants in Northern Ireland

Data analysis: This section seemed far too short. Give a little more color to it. Two sentences is not enough. 

Thank you for highlighting this, we have included additional information about the data analysis process: 

Page 12 line 324-330:

Raw data were downloaded from Microsoft Forms, cleaned and checked for duplicates, and unique participant ID numbers generated. Data were then handled using IBM SPSS Statistics (version 28.0 for Mac). Raw survey responses were coded following COVID-HL study documentation protocol, this ensures consistency across international study data. Next, descriptive statistics were calculated describing frequencies and percentages, mean values and standard deviations for the variables listed above. This was done in three phases, as a whole sample, by country and by gender. Independent samples t-tests explored differences between gender, with the level of statistical significance set as a two-sided p < 0.05.

Results

General: It seems like the health and well-being data contradicted each other (to a degree). Thus, some discussion of this is required in the discussion section. I would have assumed lower health given the other subsection scores. 

Thank you for this observation, we agree there is some contradiction regarding the reporting of general health compared to other measures of well-being and stress. This is an interesting finding as it is fair to assume this would be lower given the low well-being, high work-related stress amongst others experienced by senior leaders in this study. We have discussed this finding in relation to wider longitudinal studies, suggesting possible protective factors of SoC and higher occupation on general health. We have also acknowledged the need for further research in this area, particularly regarding the association between general health status and other variables in the study. 

Page 23-24 line 562-596:

Interestingly, senior leaders in this study reported high general health status, contradicting wider study findings regarding low well-being and high work-related stress. How the term ‘general health’ was interpreted by senior leaders is unclear, this could be perceived as encompassing physical and/or psychological dimensions. It is plausible that whilst leaders in this study report work-related stress which impacts well-being, they may engage in health-promoting behaviours (e.g. physical activity), and thus perceive their health status as high. This data was not captured and requires further study, amongst a paucity of evidence regarding the presence of health-promoting behaviours in this population. Another mechanism may be through SoC, unpublished evidence from Sweden indicates the protective influence of SoC in reducing the impact of work-related stress on general health [83]. This is further supported by longitudinal research in the UK that suggests a salutogenic effect of strong SoC on chronic disease incidence [84]. Whilst associations between variables were not explored in this study, this warrants the need for further research to examine associations between perceived general health with other well-being and work-related indicators. Furthermore, although senior leaders in this study report lower well-being that the general population, perceived general health status is comparable to Census data collected in England and Wales in 2021 [85]. This is in line with research from France demonstrating comparable general health status between educational practitioners and other professions [86]. Senior leadership is considered a high occupational grade, this may offer protection for physical health as evidenced by findings in the Whitehall and Whitehall II cohort studies regarding occupation rank and psychosocial work environment [87,88].

General: Do you have more data on the retention rates of senior leaders in Wales and Northern Ireland? If so, that would reinforce the claims you make by a lot. Tying to actual burnout levels now would be great. 

We have included some data context to discussions around attrition, including available data from the Welsh Government regarding attrition. As specific breakdown by all senior leadership positions is not publicly available, we have included stats for secondary school leaders and educational practitioners more generally, to demonstrate attrition. This data is not publicly available in Northern Ireland. The following revisions include:

Page 20 line 475-483:

Available data for Wales shows an increase in senior leaders from secondary schools leaving the profession over the last 10 years, this was highest in 2022 [24]. Whilst this data is only available for senior leaders in secondary schools, there has also been a steady increase in all teaching staff leaving the profession (e.g. n=414 in 2020, n=756 in 2022) [24]. Further, a survey of over 2,000 senior leaders across England, Wales and Northern Ireland suggests increasing dissatisfaction of school leadership as a career choice and fewer aspirations of those in deputy/assistant roles to progress to senior headship level [23]. Most recently, senior leaders in Wales and Northern Ireland have resorted to industrial action, citing workload, below-inflation pay awards and chronic school underfunding as key concerns within the profession [68,69][70]. This suggests these work-related pressures persist, and industrial action is ongoing at the time of writing [71].

We have included additional references to new announcements of ongoing industrial action in Northern Ireland. We have also added a further citation, a report by the National Association of Head Teachers (2021), see citation 71. This report surveyed over 2,000 headteachers in England, Wales and Northern Ireland and findings further support concerns regarding attrition of senior leaders, and importantly, progression to top senior leadership roles from current assistant/deputy leaders. 

Page 16: Before breaking down each category, I recommend some broad, interesting findings in a short paragraph. Give the reader a taste, then expand upon those findings in each subsection (as you do). Remember, show the forest, then the trees! This is common practice. 

Thank you for this suggestion. Due to increased word count with added revisions, we feel this is sufficiently addressed within the first paragraph of the Discussion chapter which summarises the key findings: 

Page 19 line 434-441:

This study aimed to examine the working situation, well-being and work-related stress of senior school leaders in Wales and Northern Ireland and explore gender differences during a period of educational crisis leadership within the COVID-19 pandemic between 2021 and 2022. This is part of an ongoing international study through the COVID-HL network, with specific relevance to current and emerging policy and practice in Wales and Northern Ireland. Findings suggest that senior school leaders reported high workloads, low well-being, depressive symptoms and high work-related stress. Senior leaders also had high levels of exhaustion and psychosomatic complaints indicating burnout and were engaging in self-endangering working behaviours, these were significantly higher in female senior leaders.

We feel that the majority of paragraphs begin with an overview of key findings for that outcome (e.g. see below). On occasions where the paragraph does not start with headline findings, we believe it important to maintain a flow in writing, by introducing the next discussion of findings with a linking sentence.

For example: 

Page 20 line 461-462:

The majority of senior leaders in this study reported low well-being, lower than the general adult population pre-pandemic [60] and COVID-HL international study data from Hong Kong [33,34] and Switzerland [35].

Page 19 line 469-470:

The large majority of senior leaders in this study reported a higher workload than before the pandemic and working at least 50 hours per week

Page 21 line 495-496:

In this study, senior leaders reported often giving up leisure activities in favour of work, sacrificing sufficient sleep and waiving breaks during the working day

Thank you again for your positive words and productive comments, we feel this has significantly strengthened the manuscript.

---

## [Decision Letter · Decision Letter 1]

17 Mar 2024

PONE-D-23-26570R1The well-being and work-related stress of senior school leaders in Wales and Northern Ireland during COVID-19 “educational leadership crisis”: A cross-sectional descriptive studyPLOS ONE

Dear Dr. Marchant,

Thank you for submitting your manuscript to PLOS ONE. After careful consideration, we feel that it has merit but does not fully meet PLOS ONE’s publication criteria as it currently stands. Therefore, we invite you to submit a revised version of the manuscript that addresses the points raised during the review process.

We look forward to receiving your revised manuscript.

Kind regards,

Ali B. Mahmoud, Ph.D.

Academic Editor

PLOS ONE

Journal Requirements:

Reviewers' comments:

Reviewer's Responses to Questions

**Comments to the Author**

1. If the authors have adequately addressed your comments raised in a previous round of review and you feel that this manuscript is now acceptable for publication, you may indicate that here to bypass the “Comments to the Author” section, enter your conflict of interest statement in the “Confidential to Editor” section, and submit your "Accept" recommendation.

Reviewer #1: All comments have been addressed

2. Is the manuscript technically sound, and do the data support the conclusions?

Reviewer #1: Yes

3. Has the statistical analysis been performed appropriately and rigorously? 

Reviewer #1: Yes

4. Have the authors made all data underlying the findings in their manuscript fully available?

Reviewer #1: Yes

5. Is the manuscript presented in an intelligible fashion and written in standard English?

Reviewer #1: Yes

6. Review Comments to the Author

Reviewer #1: Overall, I commend the authors. The conceptual framing is much clearer, and the paper is well organized. I only have a few more recommendations. However, well done!

Abstract

Sentence 2: this sentence read a little too long this early in the work. I would chop into 2 sentences/complete thoughts.

I would spell out ‘HL’- Health literacy; I wanted to know right away who these folks were, even if brief.

Intro

The into is very well written, but I think the purpose of the study came a little late- about five pars. in. I recommend saying your study upfront, then going into all the great detail. This creates a nice ‘sign post’ for reader, easing any concerns.

Methods

I would say right away your sample, then talk about your context. That way, we know right away. Just helps the reader not have to dig for it. In parentheses is fine in the first paragraph, then expound later (as you do).

Measure: I would say each measure first (in a few tight sentences), then talk about them as you do. Again, just helps the reader follow the progression.

-Was the SOC and WHO-5 part of the principal survey? This is a little unclear. Or are there multiple surveys used?

Results

I love how you match the measures in the methods described directly to the headers here; super helpful!

Before diving right into the results, provide a little organizer paragraph. What are you planning on telling else? What are the key takeaways? Give us a quick snapshot. Again, just about guiding the reader along your journey.

Discussion

Page 20: I would avoid the word ‘interesting’ when referring to data. Be a little more impartial on word choice.

Page 23, line 449: This sentence (another mechanism..) reads off.

Page 24, line 462: A lot of research in the US, AUS, and GER back this up. I would include/cite. Here are some examples from School Leadership & Management, an intl journal (Streipe et al., 2023; De Voto & Superfine, 2023; Beckmann & Domnique Klein, 2023)

7. PLOS authors have the option to publish the peer review history of their article (what does this mean?). If published, this will include your full peer review and any attached files.

Reviewer #1: **Yes: **Craig De Voto

---

## [Author Response · Author response to Decision Letter 1]

18 Mar 2024

Response to reviewer

We would like to express our sincere gratitude to Dr De Voto for a second round of suggested revisions, and again for the productive and positive comments regarding the manuscript. This review process has significantly improved the manuscript, and we are grateful for the time taken to provide such useful feedback. 

Please find below a bullet point list of revisions followed by changes within the manuscript addressing these, including page and line number.

Abstract

• Sentence 2: this sentence read a little too long this early in the work. I would chop into 2 sentences/complete thoughts.

Page 2 line 3:

Senior school leaders were at the forefront and were exposed to particularly high demands during a period of “crisis leadership”. This occupation were already reporting high work-related stress and large numbers leaving the profession preceding COVID-19.

• I would spell out ‘HL’- Health literacy; I wanted to know right away who these folks were, even if brief.

Page 2 line 5-6:

This cross-sectional descriptive study through the international COVID-Health Literacy

Introduction

• The into is very well written, but I think the purpose of the study came a little late- about five pars. in. I recommend saying your study upfront, then going into all the great detail. This creates a nice ‘sign post’ for reader, easing any concerns.

We have included a sentence stating the main aim of this study in the second paragraph, following an initial background to the topic. 

Page 3 line 46-50:

Senior school leaders are responsible for all aspects of school life and therefore had to cope with high demands due to the COVID-19 pandemic, a period defined as “crisis leadership” [10,11]. This study aimed to examine the working situation, well-being and work-related stress of senior school leaders in Wales and Northern Ireland during this period of crisis leadership.

Methods

• I would say right away your sample, then talk about your context. That way, we know right away. Just helps the reader not have to dig for it. In parentheses is fine in the first paragraph, then expound later (as you do).

Page 6 line 100-103:

The COVID-19 School Principals Survey was administered to senior school leaders, recognised in this study as headteachers, deputy/assistant headteachers, or school staff with leadership and/or management responsibilities. The survey was conducted in Wales (n=172) between June and November 2021 and Northern Ireland (n=151) between March and May 2022.

• Measure: I would say each measure first (in a few tight sentences), then talk about them as you do. Again, just helps the reader follow the progression.

Page 9 line 170-174:

Measures

The 33-item COVID-19 School Principals Survey captured demographic characteristics in addition to the integration of adapted or standardized scales. These included work-related factors (weekly workload/teaching load), general health, well-being (WHO-5), work-related Sense of Coherence (SoC), exhaustion and psychosomatic complaints (short-form items from the Burnout Assessment Tool (BAT)), self-endangering behaviour (self-endangering work behaviour scale) and perceived stress (Perceived Stress Scale).

• Was the SOC and WHO-5 part of the principal survey? This is a little unclear. Or are there multiple surveys used?

Thank you for drawing our attention to this, we have included a sentence addressing your previous comment regarding a brief paragraph about the measures that makes it clear that existing scales were integrated into the school principal survey. 

Page 9 line 170-174:

Measures

The 33-item COVID-19 School Principals Survey captured demographic characteristics in addition to the integration of adapted or standardized scales. These included work-related factors (weekly workload/teaching load), general health, well-being (WHO-5), work-related Sense of Coherence (SoC), exhaustion and psychosomatic complaints (short-form items from the Burnout Assessment Tool (BAT)), self-endangering behaviour (self-endangering work behaviour scale) and perceived stress (Perceived Stress Scale).

Results

• I love how you match the measures in the methods described directly to the headers here; super helpful!

Thank you! We are glad this organisation is useful for the reader.

• Before diving right into the results, provide a little organizer paragraph. What are you planning on telling else? What are the key takeaways? Give us a quick snapshot. Again, just about guiding the reader along your journey.

Thank you for this useful suggestion and those above regarding structure, we have included an introductory paragraph at the start of the results section to summarise the key findings.

Page 13 line 265-271:

Results

Findings from this study suggest that senior school leaders reported high workloads (54.22±11.30 hours/week), low well-being (65.2% n=202, mean WHO-5 40.85±21.57), depressive symptoms (WHO-5 34.8% n=108) and high work-related stress (PSS-10: 29.91±4.92). Senior leaders also had high levels of exhaustion (BAT: high/very high 89.0% n=285) and psychosomatic complaints (experiencing muscle pain 48.2% n=151) indicating burnout. Senior leaders in this study were engaging in self-endangering working behaviours (74.7% gave up leisure activities in favour of work, 63.4% sacrificed sufficient sleep), these were significantly higher in female senior leaders (work extensification males: 4.21 ± 0.62, females: 4.41 ± 0.50).

Discussion

• Page 20: I would avoid the word ‘interesting’ when referring to data. Be a little more impartial on word choice.

Page 21 line 410-411:

Furthermore, findings from the teaching profession in Switzerland reported that prolonging working hours mediates the relationship between work overload and exhaustion, thus, work extensification is used as a coping mechanism to high working demands [74].

• Page 23, line 449: This sentence (another mechanism..) reads off.

Page 24 line 476-478:

This high general health status reported by senior leaders in this study may also be explained through SoC, with unpublished evidence from Sweden indicating the protective influence of SoC in reducing the impact of work-related stress on general health [83].

• Page 24, line 462: A lot of research in the US, AUS, and GER back this up. I would include/cite. Here are some examples from School Leadership & Management, an intl journal (Streipe et al., 2023; De Voto & Superfine, 2023; Beckmann & Domnique Klein, 2023)

Thank you for sharing this interesting international evidence, this has been added as below.

Page 25 line 499-502:

International evidence has demonstrated the importance of organisational preparedness, ensuring capacities are in place across systems and settings to safeguard the well-being of those at the forefront of decision making and leadership [90]. This should be considered in times of crisis and periods of higher work-related demands requiring adaptations to roles and responsibilities [91].

---

## [Decision Letter · Decision Letter 2]

22 Mar 2024

The well-being and work-related stress of senior school leaders in Wales and Northern Ireland during COVID-19 “educational leadership crisis”: A cross-sectional descriptive study

PONE-D-23-26570R2

Dear Dr. Marchant,

We’re pleased to inform you that your manuscript has been judged scientifically suitable for publication and will be formally accepted for publication once it meets all outstanding technical requirements.

Kind regards,

Ali B. Mahmoud, Ph.D.

Academic Editor

PLOS ONE

Additional Editor Comments (optional):

Reviewers' comments:

Reviewer's Responses to Questions

**Comments to the Author**

1. If the authors have adequately addressed your comments raised in a previous round of review and you feel that this manuscript is now acceptable for publication, you may indicate that here to bypass the “Comments to the Author” section, enter your conflict of interest statement in the “Confidential to Editor” section, and submit your "Accept" recommendation.

Reviewer #1: All comments have been addressed

2. Is the manuscript technically sound, and do the data support the conclusions?

Reviewer #1: Yes

3. Has the statistical analysis been performed appropriately and rigorously? 

Reviewer #1: Yes

4. Have the authors made all data underlying the findings in their manuscript fully available?

Reviewer #1: Yes

5. Is the manuscript presented in an intelligible fashion and written in standard English?

Reviewer #1: Yes

6. Review Comments to the Author

Reviewer #1: I have no further comments. Everything was sufficiently addressed, and I happily endorse this MS for publication.

7. PLOS authors have the option to publish the peer review history of their article (what does this mean?). If published, this will include your full peer review and any attached files.

Reviewer #1: **Yes: **Craig De Voto

---

## [Editor Report · Acceptance letter]

29 Mar 2024

PONE-D-23-26570R2 

PLOS ONE

Dear Dr. Marchant, 

I'm pleased to inform you that your manuscript has been deemed suitable for publication in PLOS ONE. Congratulations! Your manuscript is now being handed over to our production team.

Kind regards, 

on behalf of

Dr. Ali B. Mahmoud 

Academic Editor

PLOS ONE